# Liquid Biopsy in Head and Neck Cancer: Its Present State and Future Role in Africa

**DOI:** 10.3390/cells12222663

**Published:** 2023-11-20

**Authors:** Dada Oluwaseyi Temilola, Henry Ademola Adeola, Johan Grobbelaar, Manogari Chetty

**Affiliations:** 1Department of Craniofacial Biology, Faculty of Dentistry, University of the Western Cape, Tygerberg Hospital, Cape Town 7505, South Africa; mchetty@uwc.ac.za; 2Department of Oral and Maxillofacial Pathology, Faculty of Dentistry, University of the Western Cape, Tygerberg Hospital, Cape Town 7505, South Africa; henry.adeola@uct.ac.za; 3Division of Dermatology, Department of Medicine, Faculty of Health Sciences and Groote Schuur Hospital, University of Cape Town, Cape Town 7925, South Africa; 4Division of Otorhinolaryngology, Department of Surgical Sciences, Faculty of Medicine and Health Sciences, Stellenbosch University, Tygerberg Hospital, Cape Town 7505, South Africa; jgrobbelaar@sun.ac.za

**Keywords:** Africa, liquid biopsy, head and neck cancer, circulating tumor cells, cell-free DNA, exosomes

## Abstract

The rising mortality and morbidity rate of head and neck cancer (HNC) in Africa has been attributed to factors such as the poor state of health infrastructures, genetics, and late presentation resulting in the delayed diagnosis of these tumors. If well harnessed, emerging molecular and omics diagnostic technologies such as liquid biopsy can potentially play a major role in optimizing the management of HNC in Africa. However, to successfully apply liquid biopsy technology in the management of HNC in Africa, factors such as genetic, socioeconomic, environmental, and cultural acceptability of the technology must be given due consideration. This review outlines the role of circulating molecules such as tumor cells, tumor DNA, tumor RNA, proteins, and exosomes, in liquid biopsy technology for the management of HNC with a focus on studies conducted in Africa. The present state and the potential opportunities for the future use of liquid biopsy technology in the effective management of HNC in resource-limited settings such as Africa is further discussed.

## 1. Introduction

According to the latest GLOBOCAN data, head and neck cancer (HNC) is the seventh most common cancer worldwide [1]. Globally, HNC accounts for 4.6% of all cancer diagnoses [2]. Africa and other low-middle income countries have the highest incidence and mortality rate due to HNC [1]. Although HNCs generally have a poor prognosis globally, with a 5-year survival rate ranging between 30% and 70%, depending on tumor location and stage [3], they have been found to have a lower 5-year survival rate of around 30% in patients of African ancestry as compared to an average 5-year survival rate of 57% in Caucasians [4,5,6].

HNC, as a collective, comprises usually squamous cell carcinomas that have their origin in the epithelial surfaces of the oral cavity, the oropharynx, the nasopharynx, hypopharynx, and the larynx [7]. Less frequently HNC can also arise in the major or minor salivary glands, the sinuses, and the muscles or nerves of the head and neck [8].

The second most common malignancy in the head and neck region are lymphomas, which are malignant neoplastic proliferations of cells of the immune system. Of the almost 600 lymph nodes in the body, approximately 300 are found in the head and neck. The mucosa-associated lymphoid tissue (MALT), which are aggregates of lymphoid tissue are also present in the tonsils and salivary glands. In Africa, patients with HIV and consequent immune suppression are at an increased risk of developing lymphomas. Often, the microorganisms associated with these lymphomas are the Epstein–Barr virus, HHV8, HTLV-1, and HCV.

Ordinarily, malignancies of the skin of the head and neck, the brain, the eye, the esophagus, and the thyroid gland are not classified as head and neck cancers. In instances when the primary tumor cannot be located in the head and neck, malignant squamous cells are still identified in the lymph nodes of the head and neck, and the disease is then referred to as metastatic squamous cell carcinoma with an occult or unknown primary.

There is evident under-reporting of HNC in Africa due to a lack of cancer registries, cancer control programmes, modern health infrastructure, access to healthcare, and finances as well as lower educational levels and the existing religious and cultural beliefs [9,10]. These are fundamental barriers to the early presentation and diagnosis of HNC among African populations [11]. The incidence of HNC is escalating in Southern and Eastern Africa attributable to increased alcohol consumption and tobacco use and traditional practices like chewing khat and tobacco, which are carcinogenic [12]. If detected early, HNC can be treated more successfully.

In Africa, the increased heterogeneous nature of HNC at the molecular level has hindered the identification of diagnostic biomarkers and the development of targeted therapeutics for this category of tumors [13]. In technologically advanced countries, the HNC genome, transcriptome, and metabolome have been scrutinized, yet these findings have not translated to an improved clinical management of these cancers on the African continent.

The subtypes of HNC are histologically and molecularly dissimilar; however, in Africa, they are treated equivalently with limited success [14,15]. The lack of biomarkers for early diagnosis and targets for personalized treatment suggests an urgent need for an improved appreciation of the complex molecular biology of HNC in Africa [16,17].

The all-encompassing term ‘Liquid biopsy’ is used to describe the analyses of bodily fluids, which include blood, urine, cerebrospinal fluid, and saliva. Liquid biopsies have lower procedural costs [18], are easily repeatable, and are more reliable [19]. Definitions of liquid biopsy within the cancer diagnostics field tend to focus on tests that target specific biomarkers [20].

Although liquid biopsies are relatively recent and not yet available for most cancer types [21], they are currently being investigated and validated as a less invasive alternative towards the diagnosis of HNC in its earliest stages [22]. They can be used, in addition to a tumor biopsy, to support the initial diagnosis of cancer, enhance our understanding of the heterogeneity of a tumor [23], and, several times throughout a treatment, to monitor the tumor and its response to therapy [24]. Liquid biopsies are also being researched to see if they can aid in the detection of cancer recurrence in patients who have previously received treatment [25].

Liquid biopsy involves the evaluation of biological markers such as released circulating tumor cells (CTCs) and circulating tumor DNA (ctDNA) in the blood or body fluid of a patient with cancer [24]. Other bio-particles that can also be characterized using liquid biopsy include circulating cell-free RNA (cfRNA), exosomes, and platelets. In addition to blood, saliva, urine, pleural effusions, seminal plasma, sputum, cerebrospinal fluid, and stool samples are physiological fluids that can be utilized for a liquid biopsy [20].

In this review, we will discuss the common circulating molecules used for liquid biopsy technology with a focus on their role in the management of HNC patients. We will also highlight liquid biopsy studies conducted in the field of HNC with a focus on Africa-based studies (Table 1). Finally, we will discuss the present state and the future role of liquid biopsy as biomarker detection method for the management of HNC among African populations.

## 2. Circulating Tumor Cell in Head and Neck Cancer

Cancer metastasis remains one major challenge of cancer management and a major cause of cancer-related deaths [44,45]. Although the scope of cancer metastasis is not fully understood, it is considered to be a multistep process involving the intravasation, extravasation, migration, and regeneration of dislodged tumor cells from primary site to distant tissues. Dislodged tumor cells, which travel through blood and lymphatic system, are referred to as circulating tumor cells (CTCs) [46].

Notably, CTCs are in a very low concentration in blood, i.e., approximately 1–100 cells per mL among a great number of red and white blood cells, making it difficult to isolate CTCs from blood samples [47]. The few numbers of CTCs in circulation are partly due to the short half-life (2 h) of CTCs in blood and the fast rate at which CTCs are eliminated through processes, such as immune attacks, shear stress, and oxidative stress [48,49]. This leads to CTCs developing different mechanisms for adapting to their harsh environment such as the epithelial-to-mesenchymal transition (EMT) [50]. The mesenchymal transition of CTCs makes them lose surface markers such as the epithelial cell adhesion molecule (EpCAM) and E-cadherin, thereby enabling them to transit through the extracellular matrix (ECM) into the microvasculature [51,52,53]. Platelets and other blood cells have also been shown to provide protection to CTCs in circulation [54]. Transforming growth factor-β (TGF-β), secreted by platelets, helps to inactivate and protect CTCs against natural killer (NK) cells [55]. A transfer of the major histocompatibility complex (MHC) I complex from granular platelets to CTCs protects them from NK cells [56].

Several techniques have been applied for isolating CTCs from blood. These techniques are either based on the enrichment of CTCs using specific tumor cell markers (label-dependent) or based on a physical property of tumor cells (label-independent). The CellSearch (Verdex LLC, San Diego, CA, USA), which is presently the only US Food and Drug Administration (FDA)-approved method for CTCs, is based on the expression of epithelial cell adhesion molecules (EpCAM) on tumor cells [57]. CellSearch has limited applications due to the lack of expression of the surface marker, EpCAM, on some CTCs [58]. There are many other label-dependent techniques for detecting CTCs, such as the EPithelial ImmunoSPOT (EPISPOT) assay, which makes use of antibodies to detect the protein markers of CTCs, and AdnaTest detection system (Qiagen, Hilden, Germany), which combines immuno-magnetic separation and multiplex RT-PCR for the detection and isolation of CTCs.

CellSearch and other label-dependent techniques have limited applications due to the lack of expression of the surface markers such as EpCAM [58]. Also, there is loss of some surface markers present on CTCs as they undergo epithelial-to-mesenchymal transition (EMT) [58,59]. These limitations may make the label-independent techniques more efficient in the enrichment and isolation of all subtypes of CTCs including the mesenchymal phenotypes. Some of the CTC’s size and density-based techniques include Parsortix isolation system, Isolation by Size of Epithelial Tumor cells (ISET), FaCTChecker, and Oncoquick platforms [60,61,62,63]. Dielectrophoresis (DEP), which is a recent method used for CTC isolation to separate CTCs based on their dielectric properties, is a relatively new and continually evolving method for isolating CTCs based on their dielectric properties [64,65].

Previous studies have described the role played by CTCs in the diagnosis, prognosis, and treatment monitoring of HNC. The detection rate of CTCs in HNC has been shown to vary from 16 to 80% depending on the detection method used and the cancer site [66,67,68,69]. Buglione et al. reported a variation in the rate of CTC detection in different anatomical sites in the head and neck [70]. The study found a higher CTC positivity as high as 50% in oropharyngeal, hypopharyngeal, and paranasal sinus cancer than other sites in the head and neck region [70].

The diagnostic ability of CTCs in HNC has been focused more on determining the stage of the disease rather than on developing an early diagnosis tool. A study by Hristozova et al., which used flow cytometry for analyzing CTCs, found a significant correlation between the nodal stage and the CTC count [68]. Another study by Kawada et al. used a micro-filter (CellSieve) to detect CTCs and reported a significant relationship between the CTC count in both the advanced stages of the disease (III–IV) and the T status (T3–4) [71].

Several studies have explored the prognostic ability of CTCs in HNC [72,73,74,75]. Jatana et al. were the first to report a significant correlation between the CTC presence and the worse clinical outcome [69]. By using a negative depletion method, they found that the presence of CTC correlated with a decreased disease-free survival (DFS), and patients having more than 25 CTCs/mL had a poor clinical outcome [69]. Grisanti et al., using the CellSearch^®^ system, showed that a baseline of more than two CTC counts was prognostic for a worse patient’s overall survival [74]. A recent study by Liu et al. found an anatomical difference in the prognostic ability of CTCs in HNC [75]. The study reported CTCs to have a better prognostic ability in hypopharyngeal squamous cell carcinoma than nasopharyngeal squamous cell carcinoma [75]. The prognostic role of proteins expressed on CTCs has also been explored. A study by Strati et al. reported that HNSCC patients with PD-L1 overexpression of CTCs after chemotherapy had worse progression-free survival and overall survival [76]. The study concluded that the detection of CTCs overexpressing PD-L1 may serve as an important prognostic tool for HNSCC [76]. Another study by Chikamatsu et al. reported the detection of CD47, PD-L1, and PD-L2 in HNSCC patients with recurrent and/or distant metastasis, suggesting its prognostic role in the management of HNC [77]. The role of CTC in the treatment monitoring of HNC patients has been reported in many studies. Both Hristozova and Kawada were able to show a significant decrease in the CTC count immediately post-treatment, demonstrating the potential utility of this assay in post-treatment monitoring [68,71].

There is paucity of Africa-based studies on the role of CTC as a biomarker of HNC (Table 1). A study by Moussa et al. applied RT-PCR in detecting circulating tumor cells in the blood of nasopharyngeal carcinoma (NPC) patients in the Tunisian population [26]. The study found that cytokeratin 19 RT-PCR nested assay can serve as a biomarker for the detection of minimal metastatic NPC in peripheral blood [26]. This study is the only known Africa-based study on the role of CTC as a biomarker of HNC, making it important for other studies or trials to be conducted in order to maximize the role of CTC as a biomarker for the management of HNC in Africa.

## 3. Circulating Tumor DNA in Head and Neck Cancer

Fragmented cell-free DNA (cfDNA) was first identified by Mandel in human blood in 1948 [78]. Two decades later, studies found an increased blood level of cfDNA in human diseases such as systemic lupus erythematosus, leukemia, and rheumatic arthritis, when compared with healthy individuals [79,80]. In 1977, Leon et al. described the role of cfDNA in cancer diagnosis. They found a significantly higher blood level of cfDNA in cancer patients than in healthy individuals [81]. About 10 years later, Stroun et al. found that a part of the cfDNA in cancer patients is from tumor cells and contains tumor-specific molecular alterations [82]. These earlier findings led to a series of studies on the role of cfDNA in liquid biopsy for cancer diagnosis.

CfDNA has been described in blood and other body fluids such as saliva, urine, cerebrospinal fluid, pleural fluid, and breast milk [83,84,85]. Cancer patients have been shown to have a higher concentration of cfDNA in circulation [86,87]. The higher concentration of cfDNA in cancer patients has been partly attributed to the presence of tumor-derived cfDNA, commonly referred to as circulating tumor DNA (ctDNA) [88].

The mechanism by which cfDNA is released into circulation is not fully known, although apoptosis, necrosis, and active cellular secretion are considered as the major sources of cfDNA [89]. The fragment size of cfDNA measures between 150 and 200 bp with a peak size of 166 bp [90,91]. The fragment size of ctDNA in relation to cfDNA is undetermined, as some studies showed that ctDNA is longer than cfDNA [92,93], while other studies reported that ctDNA is shorter than cfDNA [94,95]. The different mechanisms of fragmentation of tumor DNA might be responsible for the differences in the size of ctDNA.

CfDNA fragments possess the ability of entering neighboring or distal cells and alter the biology of these cells [96,97]. Also, cfDNA released in cancer patients has been shown to possess tumor-specific aberrations, such as mutations in tumor suppressors and oncogenes, microsatellite instability, DNA methylation, loss of heterozygosity, and copy number variation [98,99,100]. These findings suggest cfDNA as an important tool for cancer management.

The half-life of cfDNA in circulation is estimated to range between 16 min and 2 h [101,102]. The short half-life implies that cfDNA analysis can be used to monitor real-time changes in tumor genetic landscape. The mechanism of cfDNA clearance is not fully understood. However, cfDNA is considered to be cleared from circulation through nuclease action, which is followed by renal excretion via urine or an uptake by the liver and spleen and which is then followed by macrophagocytosis [101,103,104,105].

Studies have described the role of cfDNA in plasma and saliva as a potential biomarker of HNC [106,107,108,109,110]. Previous studies reported an increase in cfDNA concentration in HNC in comparison with that of healthy individuals, suggesting its potential role in the diagnosis of HNC [107,111]. A study by Lin et al. showed a significantly higher concentration of cfDNA in plasma of oral squamous cell carcinoma (OSCC) patients when compared with healthy individuals [107]. Another study by Mazurek et al. reported a higher concentration of cfDNA in the plasma of HNC patients when compared with healthy individuals [112]. The study further found a higher concentration of cfDNA in oropharyngeal cancer patients when compared to other HNC sites [112].

Studies have also explored the role of somatic gene mutations and DNA methylation in cfDNA as biomarkers of HNC [113,114,115,116]. A study by Wang et al. reported tissue somatic mutations in TP53, PIK3CA, CDKN2A, FBXW7, HRAS, and NRAS and are found in cfDNA obtained from plasma and saliva of HNC patients [117]. A recent study by Shanmugam et al., using targeted sequencing approach, found somatic mutations in CASP8, PIK3CA, FAT1, CDKN2A, NOTCH1, HRAS, and TP53 genes in the tissue and saliva of HNC patients [114]. The study further showed that saliva revealed more somatic mutations than a tumor, which validated the fact that cfDNA is a better representative of intratumor heterogeneity in HNC [114]. A DNA methylation study by Schröck et al. found that cfDNA from HNC patients have a significantly higher methylation level of SEPT9 and SHOX2 than that of the healthy controls [118]. The study further reported that a combination of both genes has a high specificity with a diagnostic accuracy of 0.80 [118].

Despite the growing number of studies on the use of cfDNA as a biomarker of HNC, Africa-based studies on this topic are sparse with the majority of available studies focusing only on North Africa (Table 1). Most of the few studies conducted in Africa were focused on investigating the diagnostic role of cfDNA as a biomarker of HNC (Table 1). A study by Gihbid et al., which explored the prognostic role of cfDNA, found that cell-free Epstein–Barr virus (EBV) DNA can serve as a prognostic biomarker for NPC patients in Morocco [30]. Another study by Hassen et al. also found that cell-free Epstein–Barr virus DNA can serve as a predictive biomarker of disease progression and survival among NPC patients in Tunisia [33]. Although the findings from these Africa-based studies are steps towards the potential use of cfDNA as a biomarker for HNC patients, there is a need for more studies to be conducted in larger cohorts to validate these findings. It is also imperative that studies are conducted to explore the role of cfDNA as a biomarker for HNC management in other parts of Africa, since most of the available studies were conducted in North Africa.

## 4. Circulating Tumor RNA in Head and Neck Cancer

Most studies on circulating tumor RNA are focused on RNA present in extracellular vesicles. However, studies have shown that circulating RNA such as free miRNA, mRNA, and lncRNA can be found in blood and other fluids [119,120]. These circulating RNA can serve as biomarkers in the diagnosis, prognosis, and treatment monitoring of cancers including HNC.

Circulating RNAs are released into circulation through processes such as necrosis, apoptosis, and active secretion [121]. They are produced by a regulated cleavage process and play a vital role in cell physiology and intercellular communication [122,123].

Circulating RNA found in circulation are either coding RNA or non-coding RNA, which are further classified based on the length of their nucleotide into small or long non-coding RNAs. Messenger RNAs (mRNAs) are the protein-coding regions that contain the information generated through DNA transcription in the nucleus. mRNAs are transported from the nucleus into the cytoplasm where proteins are formed as a result of mRNA translation. Long non-coding RNAs (lncRNAs) are the non-coding RNA that are longer than 200 nucleotides and do not code for any protein [124]. The lncRNAs regulate various cellular processes including cell differentiation and proliferation, thereby promoting carcinogenesis and cancer progression [124]. The most studied of the short non-coding RNA are the microRNA. They are about 18–24 nucleotide long and play a vital role in regulating gene expression at the post-transcriptional level [120].

Previous studies have explored the role of circulating RNAs as biomarkers of HNC. A study by Liu et el. found a significant increase in plasma miR31 in OSCC patients and a significantly lower level of miR31 in them after surgery, which suggests its potential use as a biomarker of OSCC [125]. A study by Li et al. described the potential role of mRNA expression patterns in the saliva of OSCC patients as an early diagnostic biomarker of OSCC [126]. A study by Fayda et al. found an association between a high plasma level of GAS5 lncRNA and disease persistence among HNC patients [127]. The study proposed a potential use of GAS5 lncRNA as a predictive biomarker for treatment response among HNC patients [127].

There is no known Africa-based study on the role of circulating RNA as a biomarker of HNC. This further shows the research gap in Africa in the search for circulating molecules, especially RNA, as potential biomarkers of HNC. The lack of Africa-based studies on circulating RNA also reiterates the urgent need for Africa-based studies to explore the role of circulating RNA as a biomarker for HNC management.

## 5. Exosomes in Head and Neck Cancer

Extracellular vesicles are lipid membrane-bound vesicles, classified into microvesicles, apoptotic bodies, and exosomes [128]. Exosomes, which range in size from 30 to 150 nm, are the most researched of the three extracellular vesicles. Exosomes develop from the inward budding of membrane of multi-vesicular bodies (MVB) and are released from the cell into the extracellular space. Exosomes are released by both normal and cancer cells and have been described in many body fluids such as blood, saliva, urine, cerebrospinal fluid, and breast milk [129,130,131] (Figure 1).

Exosomes vesicles are known to contain bioactive molecules such as nucleic acids, proteins, and lipids, transported from donor cells to recipient cells, making them play a major role in intracellular communication, cancer migration, invasion, and metastasis [133,134].

Exosomes have been shown to play key roles as biomarkers for cancer diagnosis, prognosis, and treatment monitoring [135,136]. Studies have explored the role of plasma and salivary exosomal cargoes as biomarkers for diagnosis, prognosis, and treatment monitoring of HNC.

Previous studies reported an upregulation of miR24, miR412, miR512, miR27a, and miR494 in the exosomes of saliva of OSCC patients when compared with that of healthy individuals [137,138]. Another saliva-based study by Coon et al. showed that exosomal miR365 could serve as a potential diagnostic biomarker of OSCC [139]. A study by He et al. on the plasma exosomes of OSCC patients found that miR130a could serve as a potential biomarker for the diagnosis and prognosis of OSCC [140].

The therapeutic role of exosomal miRNA was described by Tong et al. who found that miR9-enriched exosomes can increase the radiosensitivity of human papilloma virus (HPV) + HNSCC [141]. The study suggested that miR9 may possibly be useful in the treatment for HNSCC [141].

An exosomal protein study by Li et al. described the role of mesenchymal stem cell exosome-mediated matrix metalloproteinase 1 in oral carcinogenesis [142]. The study found an increasing protein expression level of matrix metalloproteinase 1 in exosomes isolated from normal oral mucosa, oral leukoplakia, and OSCC mesenchymal stem cells [142]. The study suggests the potential role of MMP1 protein as a diagnostic biomarker of OSCC [142]. Another study by Ono et al. found that a high level of heat shock protein 90 (HSP90) is associated with the poor prognosis in OSCC patients with metastasis [143]. A study by Liu et al. reported that the detection of exosomal cyclophilin A (CYPA) assisted the differentiation of NPC patients from healthy individuals [144]. The study suggested that the exosomal CYPA could help improve diagnosis when combined with the EBV antibody test [144].

However, despite the growing potential of exosomes as biomarkers of HNC, Africa-based studies on the role of exosomes as biomarkers of HNC are sparse (Table 1). The only known Africa-based study reported the diagnostic potential of salivary miRNA134 and miRNA200a as biomarkers of OSCC in an Egyptian population [35]. The findings of this study could serve as a template for other Africa-based studies on the role of exosomal cargoes as diagnostic biomarkers of HNC. It is also imperative for studies to be conducted to explore exosomal cargoes as prognostic and treatment monitoring biomarkers for HNC patients in Africa.

## 6. Circulating Proteins and Peptides in Head and Neck Cancer

The use of proteins as biomarkers for cancer has a long history in diagnostic sciences. Protein predates most other biological molecules as a biomarker for liquid biopsy. However, most of the protein biomarkers have not been satisfactorily used for cancer diagnosis. Some of the United States of America Food and Drug Administration (FDA)-approved protein markers for cancer diagnosis include prostate-specific antigen (PSA), Pro2PSA, alpha-fetoprotein (AFP), carcinoembryonic antigen (CEA), nuclear mitotic apparatus protein (NuMA, NMP22), and cancer antigens (CA-125, CA15-3, CA19-9) [145]. Despite being approved by the FDA, most of the protein markers still have limited use due to their low specificity and sensitivity.

Although there is presently no FDA-approved protein biomarker of HNC, some clinically relevant proteins markers are presently in use to aid HNC diagnosis [146]. Recent studies have explored the role of salivary protein markers as biomarkers of HNC as reviewed by Amenabar et al. [147]. A study by Yu et al. [148] found four salivary protein panels that can potentially serve as biomarkers for the early diagnosis of OSCC.

Africa-based studies on the role of circulating protein as a biomarker of HNC are sparse (Table 1). An Egyptian study by El-Benhawy et al. reported that adipocyte-fatty acid binding protein (A-FABP) is associated with the clinical stage of HNSCC and a higher plasma level of A-FABP is a risk marker for HNC [40]. Another study conducted in Egypt found that salivary mucin1 could serve as a potential diagnostic biomarker of OSCC [36]. A study by Houali et al. showed the presence of LMP1 and BARF1 in the serum and saliva of Algerian NPC patients [42]. The study suggested that both proteins could serve as good diagnostic markers for NPC in the North African population [42].

However, the findings of these studies are limited in their applicability among African populations due to factors such as the lack of studies from other African regions and the lack of clinical trials to validate these findings. Therefore, there is a need for larger studies to confirm the validity of the proteins found in the African studies. There is also a need for studies to be conducted in other Africa regions to corroborate the findings of studies from North African countries. This may help in identifying proteins or the panel of proteins that could be used as diagnostic, prognostic, or treatment monitoring biomarkers for HNC patients in Africa.

## 7. The Present State of Liquid Biopsy for Head and Neck Cancer in Africa

The burden of oral cancer is still grossly underestimated in Africa, and the combined burden of HNC has been found to be high on the continent [149,150]. Hence, leveraging a high level of scientific innovation is important in these regions, as economic, educational, health, infrastructural, technological, and manpower developments are limited [151]. Even though poverty continues to decline in many developing countries, the converse is true for sub-Saharan Africa, with many countries having a low Gross National Income (GNI) per capita (ca. USD 4125 or less) [152,153]. Furthermore, the exponential healthcare cost in this region continues to pose an impediment to healthcare access for the indigent. Factors responsible for disparities in HNC outcomes among African patients are poorly understood, and hence, innovative improvements in early detection and evidence-based intervention guidance are crucial. Hence, there is a pressing need to improve the dismal prognosis of HNC and cancer-related costs for African economies and circumvent such unfavorable outcomes with precision techniques that focus on prevention rather than cure. There is adequate evidence that liquid biopsy techniques are of tremendous benefit in the management of HNC. The early identification and determination of prognosis of HNC still remains a global challenge, particularly on the African continent [149,154]. The high burden and heterogeneity of HNC, coupled with the paucity of precision screening tools, often puts a strain on health spending allocation and often results in poor survival outcomes in resource-constrained regions [151]. The recent advancement, the diagnostic precision, the ease of sample collection, and the non-invasiveness of liquid biopsy techniques make it an attractive option for prompt HNC detection and management in Africa [155]. Despite the appealing application opportunities for liquid biopsy utility in Africa, there remain significant challenges in the development and implementation of broad-based clinical applications of liquid biopsy technology for HNC in the post-genomic era. There has also been a trend reversal between the classic alcohol and tobacco-induced HNC and HPV-induced oropharyngeal HNC [154,156,157], with the latter on the rise, while the former is on a decline especially in western countries. The need to switch from the current invasive tissue/cytology-based HNC diagnostic gold standard to a non-invasive, real-time, liquid biopsy-based approach cannot be overemphasized to save healthcare cost and lives in a region where there is a high HNC burden and resources are scarce [149,155]. This novel diagnostic technological approach will improve patient compliance, the obviation of need for surgical biopsies, accessibility to diagnostic substrate, and the early detection and treatment monitoring of HNC in a non-invasive manner. Liquid biopsy-based biomarker identification via exosomes, droplet digital PCR (ddPCR)-dependent ctDNA, and CTCs still requires platform optimization to improve specificity, sensitivity, and standardization under various circumstances [158]. It would certainly be beneficial to begin the development of population-based precision biomarkers for African HNC patients. Although the clinical utility and standardization of liquid biopsy technology for HNC diagnosis and prognosis on the African continent are still in their infancy, liquid biopsy offers potential opportunities to transform the HNC management landscape in resource-scarce African settings [159]. The burgeoning knowledge on the genetics and molecular basis of HNC affecting African patients provides a window of opportunity for broad-based application of liquid biopsy technology on the continent [155,159]. Unfortunately, only a few studies have emphasized the importance and benefits of liquid biopsy technology in the African HNC management settings [155]. Many liquid biopsy studies have emerged from Africa but are mostly focused on cancer affecting other areas of the human anatomy such as the liver [160,161,162,163], breast [164,165,166], colon [167,168,169], lung [170,171], skin [172], stomach [173], ovary [174], bladder [175,176], and prostate glands [177,178]. Other African liquid biopsy studies focused on lymphoreticular malignancies such as non-Hodgkin lymphoma [179,180] and acute leukemia [181,182]. Most of these studies were carried out in high-income North African countries such as Egypt and Tunisia. Also, a few of these studies were carried out in South Africa, which is a leading African economy and has one of the largest per capita gross domestic product (GDP) in Africa [183,184]. Other countries in Africa from where these liquid biopsy research papers were published included Senegal, Gambia, Cameroon, and Central African Republic. A comprehensive report on key liquid biopsy cancer studies from Africa can be found in a review by Temilola et al. [159]. The critical dearth of liquid biopsy technology application and research for HNC on the African continent needs to be addressed, as African HNC is well known to harbor notable genetic polymorphisms [185]. Many of the emerging liquid biopsy products, which were tested in non-African patients, also need to be optimized for African HNC patients to maximize their benefit in a personalized manner. Epstein–Barr Virus is a canonical causative agent for undifferentiated and non-keratinizing NPC in North and East Africa, and ctDNA in liquid biopsy has been targeted for EBV latent membrane proteins such as LMP1 and LMP2 [186,187]. Also, considering that HNC cancer-causing oral habits such as cultural khat chewing is also prevalent in the Eastern Africa countries like Kenya [188], liquid biopsy technology is important for the early detection of potentially malignant oral disorders and HNC in high-risk populations. Limiting factors that may impede the clinical utility of liquid biopsy in Africa includes the lack of requisite health infrastructure, manpower, and government funding support [159]. Last but not least, the exorbitant cost for publicly available liquid biopsy service is also a deterrent for its wide use. For instance, the cost of an liquid biopsy test in Kenya is in the range of USD 7000, which is beyond the reach of the ordinary Kenya citizens [46]. These militating factors demand innovative, precise, population-based, translation research for the multiplexing of liquid biopsy analysis and solving the stage-dependent cfDNA low-frequency allele issues in HNC [189]. Also, there is a need for robust government fiscal allocation for health that can fund innovative translational dental research that will reduce the HNC burden, improve the health of the African populace, and have a meaningful sociocultural impact on African health systems. Although there is a paucity of original research articles that specifically evaluate the use of liquid biopsy technology for HNC in Africa, Adeola et al. recently reviewed the potential of practical applications of liquid biopsy-based diagnostics for the management of oral cancer in Africa [155]. In this review, the potential benefits of liquid biopsy technology for HNC in Africa and other low-and-middle-income countries were highlighted. Furthermore, key pre-analytic, analytic, and post-analytical considerations for seamless application of various liquid biopsy technologies (such as cfDNA, CTC, mRNA, miRNA, and salivary exosomes) for HNC in African patients were discussed [155]. Another identified impediment for the application of liquid biopsy for the management of HNC in Africa is the COVID-19 pandemic, which severely depleted human and financial resources on the continent and hence could further stifle the broad-based application of liquid biopsy technology for HNC on the African continent. Despite these drawbacks, liquid biopsy is poised to improve the outcome of HNC management in Africa and is likely to play a key role in combination with other adjunctive precision diagnostic tools in the era of artificial intelligence and data science.

## 8. The Future of Liquid Biopsy for Head and Neck Cancer in Africa

Despite the associated challenges with its implementation, liquid biopsy technology has been impactful in the diagnosis and real-time treatment monitoring of various cancers both within the African continent and on other continents. Hence, the future detection of HNC in Africa will potentially benefit tremendously from improving the specificity and sensitivity of liquid biopsy-based biomarkers [158]. Hence, well-designed, large-scale, multicenter, clinical trials are needed to evaluate the usefulness of various liquid biopsy approaches as neoadjuvant (or adjuvant) modalities among African HNC cohorts. Although most liquid biopsy biomarkers currently require blood as the diagnostic substrate, other non-invasive materials and body fluids such as saliva, oral rinses, exhaled breath, sputum, tear fluid, and breast milk are likely to play key roles in liquid biopsy technology in the foreseeable future. In addition, other liquid biopsy biomarkers such as exosomes, proteins, and metabolites are likely to play key roles in the detection of HNC on the African continent [190,191,192]. In light of the prevalence of infections (particularly viral infections such as HPV and EBV), liquid biopsy approaches that incorporate the investigation of the microbiome are envisaged to play a vital role in HNC management in Africa. High-throughput omics in combination with liquid biopsy technology and emerging data science tools are likely to play key roles in the identification of multimodal HNC biomarkers on the African continent. An important aspect of deploying liquid biopsy in Africa is the commitment to training and development of a clinician/dentist scientist workforce [193] to establish a wide network of dental research enterprises that can address the burden of HNC using the state-of-the-art liquid biopsy technology on the African continent [180]. For wise therapeutic decisions and the efficacious management of HNC in Africa, innovative ancillary technologies such as biosensors [194], AI-based devices, and portable electronic screening tools may need to be combined with liquid biopsy in a cost-effective manner [195,196]. The development of population screening programs with liquid biopsy technology is also imperative to maximize the benefit of liquid biopsy using a point-of-care diagnostic approach [197]. Developing a highly robust, highly reproducible, and cost-effective liquid biopsy technology in Africa requires a systematic and deliberate multi-stakeholder effort from oral healthcare providers, the African governments, and the African populace.

## 9. Conclusions

Liquid biopsy remains an interesting field of study with a great potential for causing a paradigm shift in the management of HNC and cancer in general. Most studies conducted to explore the role of liquid biopsy in HNC were conducted in America, Europe, and other developed nations, with very few conducted in Africa. This leaves a wide gap in the development of liquid biopsy assays specific for the management of HNC in Africa. In this review, we discussed the common liquid biopsy biomarkers for HNC management with a focus on the available studies conducted in Africa. We also reviewed the present state and future role of liquid biopsy in Africa, taking into consideration the prospects and challenges of its for the management of HNC in Africa. Undoubtedly, the few available Africa-based studies are a step towards the development of liquid biopsy assays specific for the African population. However, it becomes imperative for more Africa-based studies and trials to be conducted before liquid biopsy technology can be incorporated into standard HNC management in Africa.

## Figures and Tables

**Figure 1 cells-12-02663-f001:**
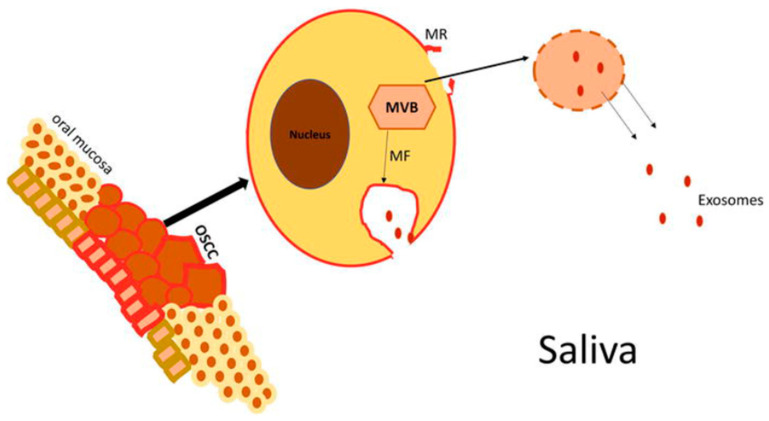
Pathways for the escape of exosomes into saliva [132]. Exosomes may be released into saliva, either by the fusion of the multivesicular body with the plasma membrane or by plasma membrane rupture and their direct release through the endosomal membrane.

**Table 1 cells-12-02663-t001:** Studies conducted in Africa to decipher the role of circulating molecules in HNC management.

Sample Type	Study Design	Detection Method	Country	Study Findings	Authors (Year)
Circulating Tumor Cell					
Blood	30 NPC patients and 20 healthy controls	RT-PCR of CK19	Tunisia	The study found an improved CK19 RT-PCR assay that is sensitive and has a high clinical specificity of 73% to detect minimal metastatic disease in patients with NPC.	Moussa et al. (2014) [26]
Circulating Tumor DNA					
Blood	40 OC patients and 52 healthy controls	NGS	Senegal	This study showed that the instability of the Bat 26 microsatellite could occur early in oral cavity cancers.	Samb et al. (2023) [27]
Blood	88 OC patients and 94 healthy controls	RT-PCR quantification of TP53 gene	Senegal	The study found a strong linkage disequilibrium between the two most common cancer-related variants in Senegalese patients.	Diaga et al. (2022) [28]
Blood	142 patients	RT-PCR for plasma EBV ctDNA	Morocco	The study reported a strong evidence that EBV DNA load testing is a promising dynamic and a minimally invasive biomarker for the prognosis and follow-up of NPC.	Gihbid et al. (2022) [29]
Blood	121 NPC patients and 60 healthy controls	RT-PCR for plasma EBV ctDNA	Morocco	The study found that pre-treatment EBV DNA can be a useful prognostic biomarker in clinical decision-making and in improving the treatment of NPC in Morocco.	Gihbid et al. (2021) [30]
Saliva	30 oral cancer patients, 21 OLP patients, 12 leukoplakia patients, and 30 healthy controls	DNA integrity index measured with RT-PCR	Egypt	The study showed that salivary DNA integrity index showed poor diagnostic abilities in differentiating between the oral cancer and premalignant lesions.	Azab et al. (2021) [31]
Blood	169 HNC patients and 261 healthy controls	SNP PCR/RFLP analysis of XRCC1, ERCC2, and ERCC3 genes	Tunisia	The study showed that XRCC1 Arg399Gln polymorphism, which correlates with occupational exposure in Tunisian population, is associated with an increased risk of developing HNC.	Khlifi et al. (2013) [32]
Blood	74 NPC patients	RT-PCR for plasma EBV ctDNA	Tunisia	The study found that the EBV DNA load quantification after treatment may be a useful predictor of disease progression and survival.	Hassen et al. (2011) [33]
Blood	64 HNSSC patients and 160 healthy controls	PCR/RFLP analysis for NAT2 gene	Tunisia	The study found the T341C mutation of NAT2 gene to be associated with an elevated risk for head and neck cancer in Tunisian population.	Gara et al. (2007) [34]
Exosomes					
Saliva	14 OSCC patients, 17 smokers, and 6 healthy controls	RT-PCR to assess the expression of microRNA-200a and microRNA-134	Egypt	The study showed that isolated salivary exosomes can provide a stable and non-invasive route for the evaluation of different salivary biomarkers, which can be a useful tool in the early detection of oral cancer.	Farag et al. (2021) [35]
Circulating Proteins and Peptides					
Saliva	40 OSCC patients and 20 healthy controls	Measurement of mucin1 expression using RT-PCR	Egypt	Th study found that the expression level of mucin1 in saliva might be a potential biomarker for diagnosing oral potentially malignant disorders and oral squamous cell carcinoma.	Abdelwhab et al. (2023) [36]
Saliva	40 OSCC patients, 40 OLP patients, and 40 healthy controls	Measurement of TF and IGFbP-3 levels using enzyme-linked immunosorbent assay	Egypt	The study showed that IGFBP-3 and Tf seem to play a role in pathogenesis of both OSCC and OLP and could be considered as reliable markers in the diagnosis of OSCC and OLP.	Elwakeel et al. (2017) [37]
Blood	17 NPC patients and 8 healthy controls	Measurement of IL-6 levels in plasma using enzyme-linked immunosorbent assay	Algeria	The study showed that the IL-6/NOS2 inflammatory signals are involved in the regulation of MMP-9- and MMP-2-dependent metastatic activity, and that high circulating nitrite levels in NPC patients may constitute a prognostic predictor for survival.	Zergoun et al. (2016) [38]
Blood	82 NPC patients and 60 healthy controls	Measurement of IGF-1 levels using enzyme-linked immunosorbent assay	Tunisia	The study found that IGF-I could serve as a good NPC diagnostic marker.	M’hamdi et al. (2016) [39]
Blood	50 HNSCC and 50 healthy controls	Measurement of plasma A-FABP levels using enzyme-linked immunosorbent assay	Egypt	The study found that a higher plasma level of A-FABP is associated with the clinical stage and risk of developing HNSCC.	El-Benhawy et al. (2016) [40]
Blood	108 NPC patients, 18 lymphoma patients, 18 autoimmune diseases patients, and 55 healthy controls	Measurement of serum EBNA1 levels using enzyme-linked immunosorbent assay	Tunisia	The study found that IgA EBNA1 ELISA may be useful for the early diagnosis and mass screening of NPC in Tunisia, even in young patients.	Ayadi et al. (2009) [41]
Blood and saliva	300 NPC patients and 50 healthy controls	Measurement of BARF1 protein and LMP1 levels using enzyme-linked immunosorbent assay	Algeria	The study showed LMP1 and BARF1 proteins to be good diagnostic markers of NPC, whereas BARF1 is a particularly promising marker for patients of all ages with NPC.	Houali et al. (2007) [42]
Blood	117 NPC and 35 healthy controls	Measurement of EBV-VCA/EAlevels using enzyme-linked immunosorbent assay	Tunisia	The study found that VCA-p18 IgA ELISA seems suitable for the routine diagnosis and early detection of NPC complications.	Karray et al. (2005) [43]

HNC—head and neck cancer, HNSCC—head and neck squamous cell carcinoma, NPC—nasopharyngeal carcinoma, OC—oral cancer, OLP—oral lichen planus, OSCC—oral squamous cell carcinoma, RT-PCR—real-time PCR, RFLP—restriction fragment length polymorphism, and SNP—single-nucleotide polymorphisms.

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
