# Peer review of "Liquid Biopsy in Head and Neck Cancer: Its Present State and Future Role in Africa"

_cells, 2023, doi:10.3390/cells12222663_

Round 1
Reviewer 1 Report
Comments and Suggestions for Authors
This review entitled “Liquid Biopsy in Head and Neck Cancer: The Present State and Future in Africa” describes in detail the studies regarding liquid biopsy in HNC patients. The study is interesting even though it is limited to only one continent.
There are some points that can be changed in order this manuscript to be more interesting for the readers.
Table 1 describes the studies from Africa regarding Liquid biopsy, but it doesn’t give any information on the outcome or the conclusions of these studies e.g. overall survival, DFS, etc
When the authors describe the knowledge regarding CTCs and HNC they refer only to studies focused on cell numbers and not on specific characteristics. There are studies for PD-L1 expression in CTCs from HNC patients. There are also other studies with more antigens expressed in CTCs with prognostic value.
In line 87 the authors use the term metabolites which is not right when they are referred to CTCs.
Section 7 is very extensive and in some cases, it repeats the same opinions and facts.
Reviewer 2 Report
Comments and Suggestions for Authors
The authors reviewed and digested comprehensive literatures about liquid biopsy for head and neck cancers (HNCs) in Africa. As the authors mentioned, liquid biopsy should be developed for early detection of HNC in resource-limited areas of the world. They explained the present state of liquid biopsy for HNC, suggesting a future load map.
Table 1 should include highlights or simple summary of the literatures. It will be more beneficial for readers.
Comments on the Quality of English Language
Line 101, HNC-Head and neck cancer, A circumflex is not necessary.
Reviewer 3 Report
Comments and Suggestions for Authors
Dear Authors,
This manuscript is mostly a review of the various approaches to carry out "liquid biopsies", and dilutes the focus on the utility of this approach in head and neck cancer (HNC) in Africa. Even when the literature on the studies is well tabulated, as in Table 1, the findings of the studies and their validity, importance and impact in the clinical setting is not discussed. In contrast Lines 155 -179 are informative. A similar pattern holds for most of the other sections, where there is an extensive review of the methods, but, less on their impact/use in HNC in Africa. Therefore, the reader's expectations of gaining insight is not met.
Round 2
Reviewer 1 Report
Comments and Suggestions for Authors
I believe the authors have successfully responded to my comments
Reviewer 3 Report
Comments and Suggestions for Authors
The manuscript is significantly improved, especially by the revision to Table 1.
Thank you.